# Development of a Highly Sensitive and Specific Monoclonal Antibody Based on Indirect Competitive Enzyme-Linked Immunosorbent Assay for the Determination of Zearalenone in Food and Feed Samples

**DOI:** 10.3390/toxins14030220

**Published:** 2022-03-17

**Authors:** Yanan Wang, Xiaofei Wang, Shuyun Wang, Hanna Fotina, Ziliang Wang

**Affiliations:** 1Henan Institute of Science and Technology, College of Animal Science and Veterinary Medicine, Xinxiang 453003, China; wyn564@126.com; 2Faculty of Veterinary Medicine, Sumy National Agrarian University, 40021 Sumy, Ukraine; 3Xinxiang Institute of Engineering, College of Bioengineering Henan, Xinxiang 453003, China; wangxiaofei1119@126.com (X.W.); wangshuyun5166@163.com (S.W.)

**Keywords:** zearalenone, immunogen, highly sensitive and specific monoclonal antibodies, indirect competitive enzyme-linked immunosorbent assay (icELISA), immunoassay

## Abstract

Zearalenone (ZEN) contamination in food and feed is prevalent and has severe effects on humans and animals post-consumption. Therefore, a sensitive, specific, rapid, and reliable method for detecting a single residue of ZEN is necessary. This study aimed to establish a highly sensitive and specific ZEN monoclonal antibody (mAb) and an indirect competitive enzyme-linked immunosorbent assay (icELISA) for the detection of ZEN residues in food and feed. The immunogen ZEN-BSA was synthesized via the amino glutaraldehyde (AGA) and amino diazotization (AD) methods and identified using 1H nuclear magnetic resonance (1H NMR), a high-resolution mass spectrometer (HRMS), and an ultraviolet spectrometer (UV). The coating antigens ZEN-OVA were synthesized via the oxime active ester (OAE), formaldehyde (FA), 1,4-butanediol diglycidyl ether (BDE), AGA, and AD methods. These methods were used to screen the best antibody/antigen combination of a heterologous icELISA. Balb/c mice were immunized with a low ZEN-BSA dose at long intervals and multiple sites. Suitable cell fusion mice and positive hybridoma cell lines were screened using a homologous indirect non-competitive ELISA (inELISA) and an icELISA. The ZEN mAbs were prepared by inducing ascites in vivo. The immunological characteristics of ZEN mAbs were then assessed. The standard curves of the icELISA for ZEN were constructed under optimal experimental conditions, and the performance of the icELISA was validated. The two ZEN-BSA immunogens (conjugation ratios, 11.6:1 (AGA) and 9.2:1 (AD)) were successfully synthesized. Four hybridoma cell lines (2B6, 4D9, 1A10, and 4G8) were filtered, of which 2B6 had the best sensitivity and specificity. The mAb 2B6-based icELISA was then developed. The limit of detection (LOD), the 50% inhibitive concentration (IC50), and the linear working range (IC20 to IC80) values of the icELISA were 0.76 μg/L, 8.69 μg/L, and 0.92–82.24 μg/L, respectively. The cross-reactivity (CR) of the icELISA with the other five analogs of ZEN was below 5%. Three samples were spiked with different concentrations of ZEN and detected using the icELISA. The average intra-assay recoveries, inter-assay recoveries, intra-assay coefficients of variations (CVs), and inter-assay CVs were 93.48–99.48%, 94.18–96.13%, 12.55–12.98%, and 12.53–13.58%, respectively. The icELISA was used to detect ZEN in various samples. The results were confirmed using high-performance liquid chromatography/tandem mass spectrometry (HPLC-MS/MS) (correlation coefficient, 0.984). The proposed icELISA was highly sensitive, specific, rapid, and reliable for the detection of ZEN in food and feed samples.

## 1. Introduction

Humans and animals are continuously exposed to various mycotoxins, threatening human health and animal husbandry development [1]. For instance, zearalenone (6-(10′-hydroxy-6’-oxo-trans-1´-undecenyl)-β-resorcylic acid, ZEN), a mycotoxin also known as F-2 toxin, is a toxic secondary metabolite produced by certain species of the genus Fusarium, including Fusarium graminearum, F. culmorum, F. tricinctum, F. roseum, F. oxysporum, F. moniliforme, and F. semitectum [2]. ZEN mainly contaminates cereals such as corn, wheat, barley, and rice. Fusarium mainly produces ZEN and a small amount of two other mycotoxins with similar structures (α-zearalenol (α-ZEL) and β-zearalenol (β-ZEL)) under natural conditions. Furthermore, ZEN can be metabolized or reduced to its homologues, such as α-ZEL, β-ZEL, α-zearalanol (α-ZAL), β-zearalanol (β-ZAL), and zearalanone (ZON) in some animal species. The molecular structures of these homologues are shown in Figure 1 [3,4]. However, ZEN is a main toxic pollutant and can induce reproductive toxicity, genotoxicity, immunotoxicity, endocrine toxicity, and carcinogenic toxicity in animals and humans. As a result, ZEN has become the main target when monitoring the quality and safety of cereal food and feed [5,6,7]. Most countries globally have implemented maximum limits of ZEN in cereal foods and feed to minimize its risks to humans and animals. For instance, according to European Union (EU) legislation, the maximum limits of ZEN are 75 µg/kg for cereals and cereal products intended for human consumption, and 100 μg/kg in compound feeds for piglets and young sows [8]. The maximum limits of ZEN are 100 μg/kg in cereals and cereal products in Italy [9] and 50 μg/kg in Australia [10]. China has set maximum limits of ZEN at 500 μg/kg in feeds and μg/kg in cereals and cereal products [11]. Therefore, establishing a single ZEN detection method with high specificity, sensitivity, and accuracy is vital for ensuring feed and food safety.

Currently, physicochemical analyses, including thin-layer chromatography (TLC) [12], high-performance liquid chromatography (HPLC) [13], gas chromatography-mass spectrometry (GC-MS) [14], and high-performance liquid chromatography/tandem mass spectrometry (HPLC-MS/MS) [15], are the main detection and validation methods for ZEN. However, these techniques are unfavorable for screening several samples, as they require expensive instruments, highly trained personnel, and complex sample pretreatments [16]. Immunoassays based on antigen-antibody reactions have been widely used to detect mycotoxin. An enzyme-linked immunosorbent assay (ELISA), as a typical representative of immunoassays, is suitable for the quantitative detection of several samples because it is fast, sensitive, accurate, and easy to operate; it also has fewer requirements for assessing sample purity [17,18].

Notably, high-quality monoclonal antibodies (mAbs) are essential to an ELISA, since their efficacy depends on the affinity and specificity of the mAbs. The production of high-affinity and highly specific mAbs depends on the design of hapten molecules and immunogen synthesis [19]. ZEN is a resorcyclic acid lactone, mainly containing 2,4-dihydroxybenzene and macrocyclic lactone rings. ZEN and its five homologues are structurally similar, except for the carbonyl or hydroxyl groups at the C6′ position and the single or double bonds at the C1′-C2′ position in the macrocyclic lactone ring. As a result, it is difficult to prepare highly specific mAbs of ZEN, due to the unique chemical properties of ZEN and its homologues [20]. In recent years, some scholars have achieved good progress in developing mAbs with high affinity and specificity for ZEN. There are several strategies for developing ZEN immunogens, including the oxime active ester (OAE), the formaldehyde (FA), the 1,4-butanediol diglycidyl ether (BDE), and the amino glutaraldehyde (AGA) methods. However, none of these methods can produce mAbs with both high affinity and specificity for ZEN. These methods develop mAbs with either serious cross-reactivity (CR) problems or low sensitivity, and thus do not meet the actual needs of single-residue detection of ZEN [20,21,22,23,24,25]. Although the AGA method can develop mAbs with high specificity for ZEN, their sensitivity needs further improvement. Therefore, the AGA method provides a valuable reference for preparing mAbs with both high affinity and specificity for ZEN [26].

In this study, the C5 position of the benzene ring of the ZEN molecule was selected as the active site for the synthesis of 5-aminozearalenone (5-NH_2_-ZEN) as a hapten via nitration and reduction reactions. Two new immunogens (ZEN and bovine serum albumin (BSA) conjugates ZEN-BSA) were obtained via AGA and amino diazotization (AD) methods. ZEN mAbs with high affinity and specificity were prepared via animal immunization with a small dose at long intervals (four weeks) and multiple-site subcutaneous injections, as well as a positive hybridioma screening technique of a heterologous indirect competitive enzyme-linked immunosorbent assay (icELISA). Finally, an icELISA with high specificity, high sensitivity, and high accuracy was established, based on ZEN mAbs, to detect single residues of ZEN in foods and feeds. Its feasibility was validated using HPLC-MS/MS.

## 2. Results and Discussion

### 2.1. Characterization of Haptens and Conjugates

The specificity of an antibody depends on the recognition of immunogen determinants by immune cells. The recognition by immune cells is closely related to the spatial conformation and the characteristic structure exposure of a hapten on the carrier protein. Therefore, the main characteristic structure of the hapten must be exposed as a part of the immunogen to the maximum extent during hapten design and immunogen synthesis to obtain the desired target antibody. The rational hapten molecular design and immunogen synthesis are critical steps in the preparation of highly specific antibodies [25,27]. The main factors affecting the characteristics of the antibody produced via the hapten design and immunogen synthesis method include the complexity of the hapten’s molecular structure, the selection of the active site, the length and structure of the introduced spacer, and the molecular binding ratio of the hapten to the carrier protein. However, the selection of active sites and the introduction of spacer arms have significant effects because different active sites may lead to different antigenic determinants of exposure, thereby affecting antibody specificity [28]. Spacer arms that are too long or too short are not suitable for the full exposure of the characteristic structure of the hapten, thereby affecting the recognition by immune cells [29]. As a result, the C5 position of the benzene ring of the ZEN molecule was selected as the active site for the synthesis of 5-NH_2_-ZEN as the hapten via nitration and reduction reaction based on the comparison and analysis of the ZEN hapten’s molecular design, the immunogen synthesis, and the characteristics of the prepared specific antibodies reported in the previous literature (Table 1) and in our previous research reports [30]. The AGA and AD methods were used to synthesize two new immunogens by introducing a five-carbon chain structure and an unsaturated N=N structure as spacer arm, respectively.

The 1H nuclear magnetic resonance (1H NMR) and high-resolution mass spectrometer (HRMS) data demonstrated that the hapten was successfully synthesized. The ultraviolet spectrometer (UV) was used to qualitatively analyze conjugates of immunogens and coating antigens. The UV spectra of BSA, ZEN, ZEN-BSA (AGA), and ZEN-BSA (AD) are shown in Figure 2a. At UV 220–360 nm, BSA had a characteristic absorption peak at 278 nm, while ZEN had characteristic peaks at 236, 274, and 316 nm. The immunogens ZEN-BSA prepared using the AGA and AD methods both showed the characteristic absorption peaks of BSA and ZEN, or the characteristic absorption peaks were shifted, indicating that the immunogens were successfully synthesized. The calculated molar ratio of ZEN to BSA was 11.6:1 for AGA and 9.2:1 for AD. The five conjugates of ZEN and ovalbumin (ZEN-OVA) prepared via the OAE, FA, BDE, AGA, and AD methods are shown in Figure 2b.

### 2.2. Preparation and Characterization of ZEN mAbs

The titers, the 50% inhibitive concentration (IC50) values and the CR values of ZEN polyclonal antibodies (pAbs) were detected using a homologous indirect non-competitive ELISA (inELISA) and an icELISA, respectively. For the ZEN-BSA (AGA) group, mouse 5 had the highest titer of 1:(6.4 × 10^3^) (Figure 3a), the lowest IC50 value of 18.77 μg/L (Figure 3b). For the ZEN-BSA (AD) group, mouse 2 had the highest titer of 1:(3.2 × 10^3^) (Figure 4a), the lowest IC50 value of 24.46 μg/L (Figure 4b). Besides, in terms of its CR values, mouse 5 had the smallest CR value of less than 5% (Figure 5a) for the ZEN-BSA (AGA) group, while mouse 2 had the smallest CR values of less than 10% (Figure 5b) for the ZEN-BSA (AD) group. As a result, mouse 5 in the ZEN-BSA (AGA) group and mouse 2 in the ZEN-BSA (AD) group were used for further cell fusion experiments.

After cell fusion culture, the positive hybridoma cells (2B6 and 4D9 derived from ZEN-BSA (AGA) and 1A10 and 4G8 derived from ZEN-BSA (AD)) were screened using three subclones. The cells were then investigated using a homologous inELISA and an icELISA. The subtypes, affinity, and stability of the four mAbs were also characterized (Figure 6). A mouse mAb isotyping kit showed that the 2B6, 4D9, and 1A10 mAbs belonged to the IgG1 isotype with a kappa light chain, while the 4G8 mAb was an IgG2a isotype with a lambda light chain (Figure 6a). The Ka of mAbs 2B6, 4D9, 1A10, and 4G8 were 7.69 × 10^9^, 4.95 × 10^9^, 3.55 × 10^9^, and 2.45 × 10^9^ L/moL, respectively (Figure 6b). The absorption and IC50 values of cell culture supernatants before cryopreservation of the hybridoma cells were similar to the values obtained after five cryopreservations, resuscitations, and subcultures, indicating that these hybridoma cells could stably secrete antibodies (Figure 6c,d). The IC50 and CR values of the four mAbs are shown in Table 1. The IC50 values of the 2B6, 4D9, 1A10 and 4G8 mAbs were 10.38, 17.23, 19.87, and 27.05 μg/L, respectively, while their CR values were 1.52–4.27%, 1.632–4.88%, 6.71–19.12%, and 7.12–20.36%, respectively. The mAb 2B6 had the lowest IC50 value and smallest CR value, and thus was selected for subsequent studies. 

The sensitivity and specificity of antibodies are two main indicators to evaluate the quality of antibodies, because they determine the performance of the established immunoassay method. Sensitivity reflects the binding ability of an antibody to a corresponding antigen or hapten, while specificity reflects the selective ability of an antibody to recognize paired antigens or haptens. A comparison between mAb 2B6 obtained in this study and other ZEN mAbs reported in previous studies is shown in Table 2. The sensitivity of mAb 2B6 was substantially lower than that of mAb 4A3-F9, which was synthesized by Sun et al. [20] via the BDE method, mAb 6C2, which was synthesized by Dong et al. [21], mAb 2D8, which was synthesized by Burmistrova et al. [23], and mAb 7B2, which was synthesized by Liu et al. [22] via the OAE method. However, the sensitivity of mAb 2B6 was higher than that of the mAb# synthesized by Gao et al. [25] via the FA method. In contrast, the specificity of mAb 2B6 was significantly better than that of the above mAbs. Moreover, the IC50 value of mAb 2B6 was lower than that of mAb 7-1-144 prepared by Teshima et al. [26] via the AGA method. The CR values of mAb 2B6 were slightly higher than that of mAb 7-1-144. Due to the high similarity in the molecular structure of ZEN and its five homologs, the prepared ZEN antibodies are difficult to distinguish [20]. Teshima et al. [26] showed that the AGA method introduces an active amino group at the C5 position of ZEN through a nitration-reduction reaction, thus exposing the main characteristic structures of ZEN, including the 2,4-dihydroxybenzene ring, a macrolide ring, and a carbonyl group at the C6’ position for easier recognition by immune cells. The antibodies produced via the AGA method are highly specific to ZEN. However, it is unclear why immune cells cannot recognize the single and double bonds at the C1’-C2’ positions and the hydroxyl group at the C6’ position. In this study, the AD method was used to synthesize ZEN immunogen to prepare highly specific antibodies for ZEN. Although the sensitivity and specificity of the prepared mAb 1A10 and 4G8 were lower than that of mAb 2B6, they were better than those prepared via the OAE, FA, and BDE methods. The AD method was inferior to the AGA method, probably because the AGA method introduced a long-chain five-carbon spacer arm, which could fully expose the characteristic structure of ZEN. In contrast, the AD method introduced a short-chain N=N spacer arm, which could not fully expose the characteristic structure of ZEN.

### 2.3. icELISA Optimization and Establishment of the icELISA Standard Curve

The type of coating antigen, the optimal working concentration of the antigen, antibody, and GaMIgG-HRP, the organic solvent concentration, the ionic strength, and the pH value affect the establishment of the icELISA method. In this study, the icELISA conditions were optimized to improve its sensitivity. The IC50 value of ZEN mAb 2B6 was determined using the homologous icELISA format and four heterologous icELISA formats. The combination with the lowest IC50 value was selected to establish the icELISA method. The results showed that the combination of mAb 2B6 (ZEN-BSA (AGA)) and the coating antigen (ZEN-OVA (FA)) had the best effect, and mAb 2B6 had the lowest IC50 value (8.69 μg/L) for ZEN (16.28% higher than that of the homologous icELISA (10.38 μg/L)). However, the other three heterologous icELISA combinations were not as effective as the homologous icELISA combinations (Figure 7a). The chessboard titration results indicated that the optimal concentrations of the coating antigen, mAb 2B6, and GaMIgG-HRP were 2 μg/mL, 0.5 μg/mL (1:10,000), and 0.6 μg/mL (1:2000), respectively (data not shown). 

Three points should be noted when checkerboard titration is used to determine the optimal working concentrations of the three optimal concentrations for the icELISA. The first point is based on the absorbance value of 1.0; the second is that positive samples are judged on the basis of positive value/negative value ≥2.1; the third is the sensitivity of the antigen-antibody reaction with different dilutions, i.e., the coloration gradient is obvious or the absorbance value is obviously different.

Methanol is an organic co-solvent widely used in an ELISA to extract the mycotoxins from the food and feed matrix. In addition to enhancing ELISA sensitivity, an appropriate methanol level helps to dissolve analytes. In this study, the Amax and the Amax/IC50 values decreased when methanol concentration exceeded 30%. However, these values were not significantly different when the methanol content was below 30% (Figure 7b). The effect of phosphate buffer saline (PBS, 0.01 M, pH 7.4) ionic strength on the icELISA is shown in Figure 7c. The Amax/IC50 value gradually decreased with a continuous increase in ionic strength, while the IC50 value gradually increased, i.e., the sensitivity of the icELISA decreased. Therefore, PBS was the best buffer, and there was no need to increase the ionic strength. The effects of the pH value on the icELISA are shown in Figure 7d. The pH values between 5.0 and 9.0 did not significantly affect the Amax/IC50 and Amax values. However, the Amax/IC50 and Amax values were highest at a pH of 7.4, indicating full binding between the antibodies and the antigen. Therefore, a PBS with a pH value between 7.2 and 7.4 was the most suitable for an icELISA system. The standard curve of the icELISA for ZEN was established based on the above-optimized conditions (Figure 8). The derived linear regression equation and the IC50 value were y = −30.728x + 78.864 and 8.69 μg/L, respectively.

### 2.4. Validation of the icELISA

Sensitivity is one of the key indicators for assessing the antigen–antibody reaction mode, i.e., the immunoassay method, represented by the limit of detection (LOD) value, the IC50 value, and the linear working range. In this study, the LOD, IC50, and linear working range (IC20 to IC80) values were 0.76, 8.69, and 0.92–82.24 μg/L, respectively, based on the results of six repeated determinations of 20 different blank samples and the above regression equation. The accuracy and precision obtained from each spiked sample measured using the optimized assay are shown in Table 3. The average intra-assay recoveries of maize meal, wheat meal, and pig feed samples were 95.95%, 93.48%, and 99.48%, respectively, while the average inter-assay recoveries of these samples were 96.05%, 94.18%, and 96.13%, respectively, indicating that the icELISA had high accuracy. The average intra-assay coefficients of variations (CVs) of the samples were 12.78%, 12.55%, and 12.98%, respectively, while the average inter-assay CVs were 12.53%, 12.78, and 13.58%, respectively, indicating that the icELISA had high precision. The comparison results of 18 actual samples and four known positive samples detected by the icELISA and HPLC-MS/MS are shown in Table 4. Six of the 18 actual samples were detected (positive rate, 33.33%). All of the four known positive samples were detected, consistent with the detection results of HPLC-MS/MS (coincidence rate, 100%). The icELISA and HPLC-MS/MS positive values of ten positive samples were 5.62–71.54 μg/L, and 5.45 to 71.69 μg/L, respectively. The CVs and the coefficient correlation of the results between the icELISA and ICP-OES were 9.8–11.2% and 0.984, respectively, indicating that the developed icELISA was reliable for the detection of ZEN in actual samples.

## 3. Conclusions 

In summary, highly sensitive and specific antibodies are essential for the detection of ZEN residues. In this study, the AGA and AD methods were used to synthesize two ZEN-BSA immunogens with a conjugation ratio of ZEN to BSA at 11.6:1 (AGA) and 9.2:1 (AD), respectively. Balb/c mice were immunized with a low ZEN-BSA dose at long intervals and at multiple sites. A homologous inELISA and an icELISA were used to filtrate the desired positive hybridoma cell lines. Four hybridoma cell lines (2B6, 4D9, 1A10, and 4G8) were filtered, of which mAb 2B6 had the best sensitivity and specificity. A heterologous icELISA based on mAb 2B6 was developed. Its LOD, IC50 value, and linear working range values at optimal assay conditions were 0.76 μg/L, 8.69 μg/L, and 0.92–82.24 μg/L, respectively. The CR values of the icELISA with five analogs of ZEN were below 5%. Three samples were fortified with ZEN and determined using the icELISA, and the average recoveries and CVs were reasonable. The contents of ZEN in actual samples determined by the icELISA and HPLC-MS/MS had a high correlation coefficient of 0.984. These results show that the proposed icELISA could provide a highly sensitive, specific, rapid, and reliable analytical method for ZEN determination in food and feed samples. 

## 4. Materials and Methods

### 4.1. Chemicals and Instruments

ZEN standard (solvent-free), α-ZAL, β-ZAL, α-ZEL, β-ZEL, and ZON standard solutions in methanol (1.0 mg/mL), aflatoxin B1 (AFB1), deoxynivalenol (DON), fumonisin B1 (FB1), ochratoxin A (OTA), T-2 toxin, polyethylene glycol 1500 (PEG 1500, 50%), hypoxanthine aminopterin thymidine (HAT), hypoxanthine thymidine (HT), 2-[4-(2-Hydroxyethyl)-1-piperazinyl] ethanesulfonic acid (HEPES), phenacetin, urea peroxide, 3,3,5,5-tetra-methylbenzidine (TMB), and Tween-20 were purchased from the Sigma-Aldrich Corporation (St. Louis, MO, United States of America (USA)). Bovine serum albumin (BSA), ovalbumin (OVA), Freund’s complete adjuvant (FCA), Freund’s incomplete adjuvant (FIA), culture media RPMI-1640 with L-glutamine, goat anti-mouse IgG conjugated with horseradish peroxidase (GaMIgG-HRP), and a mouse mAb isotyping kit were obtained from Pierce Biotechnology, Inc. (Rockford, IL, USA). Fetal bovine serum (FBS) was purchased from Hangzhou Sijiqing Biological Engineering Materials Co., Ltd. (Hangzhou, China). Glutaraldehyde (GA), 1,4-Butanediol diglycidyl ether (BDE), dimethylformamide (DMF), and dimethyl sulphoxide (DMSO) were sourced from J&K Chemicals (Shanghai, China). In addition, other reagents used are not inferior to analytical grade.

The buffers and solutions used in this study were the same as those used previously [30]. Phosphate buffer saline (PBS; 0.01 M, pH 7.4) contained 137 mM NaCl, 10 mM Na_2_HPO_4_·12H_2_O, 2.68 mM KCl, and 1.47 mM KH_2_PO_4_. The coating buffer was carbonate buffer saline (CBS; 0.05 M, pH 9.6) containing 15 mM Na_2_CO_3_ and 35 mM NaHCO_3_. The washing solution was PBS containing 0.05% (*v*/*v*) Tween-20 (PBST). The blocking buffer PBST contained 5% (*v*/*v*) swine serum. The enzyme substrate buffer was a citrate buffer (0.04 M, pH 5.5) containing 0.01% (*w*/*v*) TMB and 0.004% (*w*/*v*) H_2_O_2_. The stopping solution was 2 M H_2_SO_4_. The complete medium contained 78 mL RPMI-1640 medium, 20 mL FBS, 1.0 mL antibiotics, and 1.0 mL HEPES. The cell freezing solution was a complete medium containing 10% (*v*/*v*) DMSO. The mycotoxin standard stock solutions of ZEN, α-ZAL, β-ZAL, α-ZEL, β-ZEL, ZON, AFB1, DON, FB1, OTA, and T-2 toxin (2000 μg/mL) were prepared using 70% methanol-PBS (7:3, *v*/*v*), and diluted to a standard solution with 70% methanol-PBS (7:3, *v*/*v*). The cationized carrier protein solution of 200 mg of BSAor 135 mg of OVA was dissolved in 10 mL of PBS. Then, 7.2 mg of ethylenediamine (EDA) and 11.5 mg of EDC were sequentially added and magnetically stirred at room temperature for 2 h. The reaction solution was dialyzed using PBS at 4 °C for three days, then stored at 4 °C for subsequent use.

Cell culture plates (6, 24, and 96 wells) and culture flasks were purchased from Costar Inc. (Bethesda, MD, USA). The 96-well polystyrene microplates were purchased from the Boyang Experimental Equipment Factory (Jiangsu, China). A microtiter reader (MULTISKAN MK3, Thermo Co., Shanghai, China) was used to spectrophotometrically read the microplates. A DU-800 UV-visible spectrophotometer (Beckman Coulter Inc., Fullerton, CA, USA) was used to detect the ultraviolet (UV) spectra. The cells were cultivated in a Galaxy S-type CO_2_ incubator (RS-Biotech, Ayrshire, UK) and observed using a TS100-F inverted microscope (Nikon Company, Tokyo, Japan). A hybrid quadrupole-time of flight mass spectrometer (Q/TOF, HRMS; SYNAPT HDMS, Waters, UK) and a 700 MHz Avance III spectrometer (1H-NMR; Bruker Corporation, Billerica, MA, USA) were used to verify the chemical structure of the ZEN hapten. A 303A-1 electric heating constant temperature incubator was sourced from the Beijing Zhongxing Weiye Instrument Co., Ltd. (Beijing, China). An Exceed DZG-303A ultrapure water system was sourced from the Chengdu Kangning Special Experiment Pure Water Equipment Factory (Chengdu, China).

### 4.2. Experimental Animals and Cells

The female Balb/c mice (Licence number: SCXK (YU) 2015-0004), aged six to eight weeks and with an average weight of 20.5 ± 0.6 g, were provided by the Henan Experimental Animal Center (Zhengzhou, China). The mice were maintained in our laboratory under optimal conditions for room temperature, hygiene, and illumination, and were fed food and tap water ad libitum. All experiments were approved by the Animal Ethics Committee of the Henan Institute of Science and Technology (No. 2020HIST047) and followed the Administrative Measures on Experimental Animals of Henan Province, China. NS0 myeloma cells were provided by the Key Laboratory of Animal Immunology of the Ministry of Agriculture (Zhengzhou, China).

### 4.3. Synthesis and Identification of the 5-NH_2_-ZEN Hapten

Hapten 5-NH_2_-ZEN was synthesized from ZEN to create a reactive amino group, using the previously reported method with some modifications [26], as shown in Figure 9. Briefly: First, 0.1 mM ZEN, 0.1 mM ZrO (NO_3_)_2_, and 5 mL acetonitrile were added to a reaction tube under a nitrogen atmosphere, then stirred under a refluxed condition for 16 h. The reaction was monitored through TLC till completion. The mixture was filtrated and evaporated at reduced pressure. The crude product was purified through column chromatography to obtain a yellow solid, intermediate product 5-NO_2_-ZEN (compound b), which was confirmed using 1H NMR (1H NMR (600 MHz, CDCl_3_): δ 6.83 (1H, dd, J1 = 16.2, J2 =2.4 Hz, C-1′), 6.55 (1H, s, C-3), 5.18 (2H, C-2′, C-10′), 2.77–2.70 (1H, m), 2.60–2.56 (1H, m), 2.35–2.29 (1H, m), 2.22–2.10 (4H, m), 1.79–1.72 (1H, m), 1.68–1.55 (4H, m), 1.46–1.39 (1H, m), 1.36 (3H, d, J = 12 Hz, C-11′)). Second, 0.3 mM compound 2b was dissolved in 5 mL hydrochloric acid, and the iron powder was added to the solution. The reaction was conducted at room temperature for 2 h. The solution was filtered and concentrated in a vacuum. The resulting mixture was dissolved in 5 mL of double-distilled water (DDW), basified using K_2_CO_3_ until the solution pH was adjusted to 7.0, and extracted with DCM (3 × 10 mL). Finally, the organic layer was washed with brine, dried with anhydrous Na_2_SO_4_, then concentrated in vacuo to obtain light yellow solid target product 5-NH_2_-ZEN (compound c), which was verified using 1H NMR (1H NMR (600 MHz, CDCl_3_): δ 6.60 (1H, dd, J1 = 16.2, J2 = 2.4 Hz, C-1′), 6.40 (1H, C-3), 5.53–5.60 (1H, m, C-2′), 1.31 (3H, d, J = 6 Hz, C-11′)), and HRMS. HRMS (ESI) m/z calcd for C_18_H_24_NO_5_ [M + H] + 334.1649, found 334.1643.

### 4.4. Synthesis and Identification of Immunogen and Coating Antigen

The C5 position of the hapten 5-NH_2_-ZEN contains an active amino group, which can be coupled with BSA to synthesize the immunogen ZEN-BSA. In this study, the AGA and AD methods were used as previously described [26] to synthesize immunogen ZEN-BSA. The AGA method was performed as follows: 20 mg of BSA was dissolved in 1.0 mL of 0.1 M phosphate buffer (pH 6.8) based on the initial molar ratio of ZEN and BSA (100:1). Then, 5 mg of 5-NH_2_-ZEN was dissolved in 500 μL of methanol, and 60 μL of 2% GA solution was added and stirred at room temperature for 4 h. The resultant solution was added dropwise to the BSA solution and stirred at room temperature for 12 h. The reaction mixture was dialyzed against PBS for three days, and the resulting ZEN-BSA conjugate was stored at 4 °C for subsequent use. The AD method was conducted as follows: 20 mg of BSA was dissolved in 1.0 mL of PBS according to the initial molar ratio of ZEN and BSA (100:1), and precooled in an ice bath for 0.5 h. Then, 5 mg of 5-NH_2_-ZEN was dissolved in 500 μL of methanol. The pH was adjusted to 1.0 by adding 150 μL of HCl (0.5 M). The mixture was then precooled in an ice bath for 10 min. Then, 150 μL of NaNO_2_ (100 mg/mL) solution was added dropwise in the dark at 4 °C while stirring. The mixture was then incubated at 4 °C for 30 min. H_2_NO_3_S (50 mg/mL) was added until no nitrogen bubbles were observed to remove unreacted nitrous acid. The reaction solution was added dropwise to the precooled BSA solution after diazotization of 5-NH_2_-ZEN. The pH was adjusted to 8.0 using NaOH (1 M), then the mixture was incubated at 4 °C overnight. The conjugates were then dialyzed against PBS for three days in the dark. The dialysis solution was frequently changed. The synthetic routes of ZEN-BSA (AGA) and ZEN-BSA (AD) are shown in Figure 10.

The coating antigens ZEN-OVA were synthesized via the oxime active ester (OAE), formaldehyde (FA), 1,4-butanediol diglycidyl ether (BDE), AGA, and AD methods according to the molecular structure and active site of ZEN as described in previous laboratory research [27]. The chemical structures of the five synthetic coating antigens are shown in Figure 11.

The synthesis of the immunogen ZEN-BSA and the coating antigens ZEN-OVA was confirmed using UV scanning spectroscopy. The molecular binding ratios of ZEN to BSA and ZEN to OVA were calculated based on the Lambert-Beer law, A = εCL (where A means the absorbance value, ε represents the molar extinction coefficient, (constant value), C expresses the solute concentration, and L is the optical path (fixed value).

### 4.5. Preparation and Characterization of the ZEN mAb

Ten mice were randomly divided into two groups (five mice in each group). The mice were then immunized with either ZEN-BSA (AGA) or ZEN-BSA (AD) using a small immunogen dose (30 μg/head) at long intervals (four weeks) and multiple-site (four to six sites on the back) subcutaneous injections [31,32]. In the first inoculation, ZEN-BSA was dissolved in sterilized PBS and emulsified with the same amount of FCA. In the three-times-boosters follow-up immunization, ZEN-BSA was dissolved in sterilized PBS and fully emulsified with an equal amount of FIA. Two weeks after the last immunization, the tail-amputated blood was collected from each mouse and separated to obtain the antisera (ZEN pAb). The titer of ZEN pAb (immunoreactivity) was determined using a homologous inELISA via the immunogens ZEN-BSA (AGA) and ZEN-BSA (AD), corresponding to the coating antigens ZEN-OVA (AGA) and ZEN-OVA (AD), respectively. Furthermore, the IC50 (representing the sensitivity of the ZEN pAb) and the CR (representing the specificity of the ZEN pAb) were determined using a heterologous icELISA. Four days before cell fusion, the mouse with the highest ZEN pAb titer, the lowest IC50 value, and a significant CR value in each group received an intraperitoneal booster injection of 200 µg of ZEN-BSA conjugate without any adjuvant. The mice were then sacrificed, and the spleen was harvested to obtain hybridomas.

Cell fusion and screening of positive hybridoma cell lines were conducted as reported in previous literature [33,34]. Ten to fourteen days after cell fusion, a homologous inELISA and an icELISA were used to screen the positive hybridomas obtained from the supernatants. The positive hybridomas obtained after expanding the culture were subcloned thrice using the limiting dilution method. The in vivo-induced ascites method was adopted to produce several ZEN mAbs after obtaining the ZEN mAb hybridoma cell lines. The ZEN mAbs were then purified using the saturated ammonium sulfate precipitation method [35]. The subtypes of the prepared mAbs were identified using the mouse mAb isotyping kit. The affinity constant (Ka) of the ZEN mAbs was determined using the Batty saturation method [36,37], as follows: Ka = (*n* − 1)/[2(*n*[Ab’]t − [Ab]t)] (where *n* = [Ag]/[Ag’], [Ag]t and [Ag’]t indicate the concentrations of a coating antigen, and [Ab]t and [Ab’]t represent the corresponding 50% Amax value of the ZEN mAb concentration at different coating antigen concentrations). The cryopreserved hybridoma cells were resuscitated and passaged five times, once every ten days. The antibody titers and the IC50 values of ZEN in the supernatants at different passages were detected using the inELISA and the icELISA to determine the stability of the hybridoma cells secreting an antibody [38]. The IC50 values of the ZEN mAbs, representing their respective sensitivities, were determined using an icELISA [39]. The CR value of the ZEN mAbs, indicating their respective specificities, was determined using an icELISA for ZEN and its five homologs (α-ZAL, β-ZAL, α-ZEL, β-ZEL, and ZAN), and other common mycotoxins, such as AFB1, DON, FB1, OTA, and T-2 toxin. The CR value was calculated as follows: CR (%) = [IC50 (ZEN)/IC50 (competitor)] × 100% [40,41]. The ZEN mAb with the highest titer, the maximum affinity, the lowest IC50 value, and the smallest CR value was filtrated for the development of the icELISA.

### 4.6. Development and Optimization of the icELISA

The icELISA was developed as previously described [42,43]. Briefly, the coating antigen ZEN-OVA was diluted in CBS at 2 μg/mL, added to the microplate at 100 μL/well, then incubated at 37.8 °C for 2 h. The microplate was then washed thrice using PBST, and unbound active sites were blocked using 250 µL of blocking buffer in each well at 37.8 °C for 1 h or at 4 °C overnight. The microplate was washed again, and 50 μL of ZEN pAb or mAb (at an appropriate dilution) was added to each well, followed by the addition of 50 µL of serial dilutions of competitors in each well, then incubated at 37.8 °C for 30 min. After washing as described above, 50 μ of GaMIgG-HRP was added to each well, then incubated at 37.8 °C for 30 min. After washing six times, 100 µL of TMB freshly prepared solution was added to each well, then incubated at room temperature for 10 min. The reaction was stopped by adding 100 μL of 2 M H_2_SO_4_. The absorbance value was measured at 450 nm. Pre-immunization of serum and PBST were used as negative and blank controls, respectively. Each sample was incubated in duplicate in at least three independent experiments.

The optimal dilutions of five coating antigens, ZEN mAbs, and GaMIgG-HRP were determined via a checkerboard titration procedure to promote the performance of the icELISA. The well with an absorbance value of 1.0 at 450 nm was defined as the optimal working concentration for the icELISA. The ratio of Amax/IC50 (Amax represents maximal absorbance, and IC50 was calculated via regression equation) was used as the evaluation criteria. Some physicochemical parameters, such as methanol concentration (5, 10, 20, 30, 40, and 50%, *v*/*v*), ionic strength (5, 10, 20, 30, 40, and 50 mM PBS/NaCl), and pH value (5.0, 6.0, 7.0, 7.4, 8.0, and 9.0) of the assay buffer were also tested and optimized [44,45].

The ZEN standard stock solution (2000 μg/L) was diluted in 30% methanol-PBS (30:70, *v*/*v*) at various concentrations (0.33, 1.0, 3.0, 9.0, 27.0, 81.0, and 243.0.0 μg/L). A seven-point standard curve of the icELISA was drawn under the above optimized conditions by plotting the gradient concentrations (Log C) against the B/B0% values (proportion of mean absorbance value of the standards (B) divided by that of the zero standards (B0) in triplicate experiments). A four-parameter logistic regression equation was deduced, and the IC50 was calculated [46,47].

### 4.7. Validation of the icELISA

Several main performance indicators of the icELISA, including sensitivity, specificity, accuracy, precision, and reliability, were validated. The IC50, representing sensitivity, was calculated using a regression equation. The LOD, representing the lowest concentrations of analytes in the samples, was determined based on the mean value of 20 blank samples and 3-fold standard deviation (SD). The working linear range was defined as a 20–80% inhibition rate (IC20-IC80) [48,49]. The accuracy and precision were assessed via a spike-and-recovery test and the CVs, respectively. The blank samples (maize meal, wheat meal, and pig feed samples) were spiked with ZEN stock solution (2000 μg/L) with 70% methanol-PBS (7:3, *v*/*v*) to a final volume of 2.0 mL at four different concentrations (10, 20, 40, and 80 μg/L) of ZEN; the spiked samples were analyzed using the icELISA in six replicates. The recovery was calculated as follows: recovery (%) = (concentration measured/concentration spiked) × 100. The precision was evaluated via repeated analysis of the spiked samples and comparison of the CVs of the intra-assay and inter-assay. Intra-assay variation was measured using six replicates of each spiked concentration, and the inter-assay variation was based on the results of six different days [27,50]. The reliability analysis was performed by comparing the icELISA and HPLC-MS/MS analysis methods. Eighteen actual samples, including six maize meal, six wheat meal, and six pig feed samples, were obtained from a local market. Four positive samples, including two maize meal and two pig feed samples, were provided by the Feed Safety Quality Control Center of the Ministry of Agriculture. All the samples were equally divided into two groups. One group was subjected to the icELISA, and the other group was subjected to the HPLC-MS/MS. The correlation between the icELISA and HPLC-MS/MS analyses was then determined. The HPLC-MS/MS analysis method was performed as described by Monbaliu et al. [28].

### 4.8. Sample Preparation

Finely ground maize, wheat, and pig feed samples (2 g each) were put into a 50 mL centrifuge tube. Then, 10 mL of 70% methanol-PBS (7:3, *v*/*v*) was added for extraction. The mixture was vortexed for 5 min, then centrifuged at 3000× *g* for 5 min, and then transferred to a 100 mL volumetric flask to obtain a ZEN sample treatment solution. The obtained sample treatment solution was diluted using 70% methanol-PBS (7:3, *v*/*v*) for icELISA analysis [29].

### 4.9. Data Statistics and Image Processing

Origin Pro 2018 (OriginLab Corporation, Northampton, MA, USA)) and Excel software (Microsoft Corporation, Redmond, WA, USA) were used for plotting the standard curves and data analysis. ChemDraw 20.0 (PerkinElmer Informatics, Inc., Waltham, MA, USA) was used to sketch the chemical formulas.

## Figures and Tables

**Figure 1 toxins-14-00220-f001:**
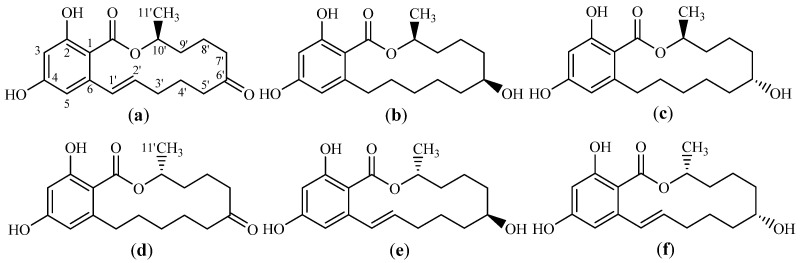
The chemical structure of zearalenone and its homologues: (**a**) zearalenone (ZEN); (**b**) alpha-zearalanol (α-ZAL); (**c**) beta-zearalanol (β-ZAL); (**d**) zearalanone (ZON); (**e**) alpha-zearalenol (α-ZEL); (**f**) beta-zearalenol (β-ZEL).

**Figure 2 toxins-14-00220-f002:**
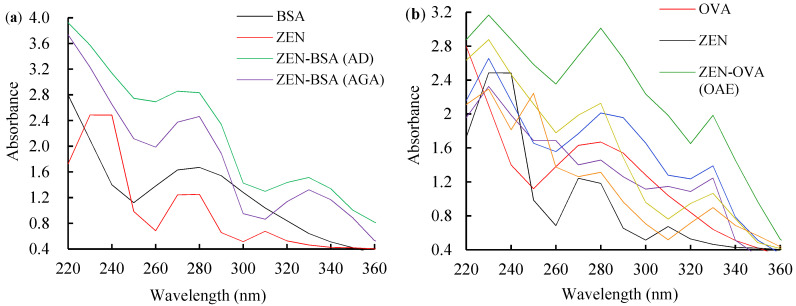
UV spectra of the ZEN-BSA and ZEN-OVA: (**a**) UV spectra of ZEN-BSA synthesized via AGA and AD methods; (**b**) UV spectra of ZEN-OVA synthesized via OAE, FA, BDE, AGA, and AD methods.

**Figure 3 toxins-14-00220-f003:**
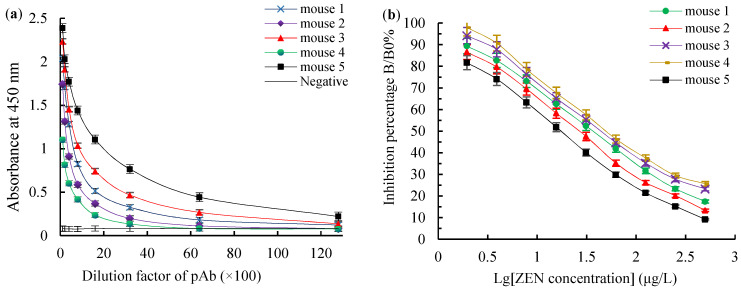
The titers and IC50 values for ZEN pAb derived from ZEN-BSA (AGA): (**a**) the titers for ZEN pAb; (**b**) the IC50 values for ZEN pAb. The values are the mean of three independent assays (*n* = 3).

**Figure 4 toxins-14-00220-f004:**
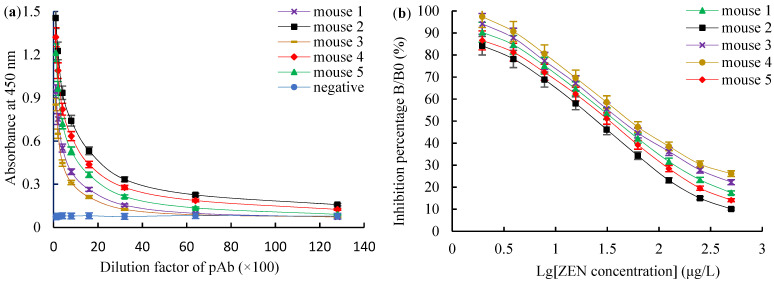
The titers and IC50 values for ZEN pAb derived from ZEN-BSA (AD): (**a**) the titers for ZEN pAb; (**b**) the IC50 values for ZEN pAb. The values are the mean of three independent assays (*n* = 3).

**Figure 5 toxins-14-00220-f005:**
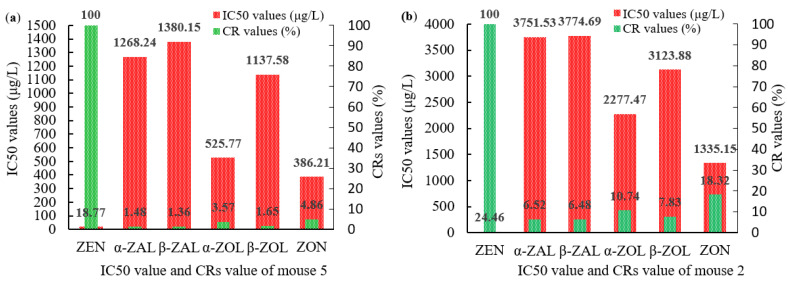
The IC50 and CR values of ZEN pAbs for mouse 5 and mouse 2: (**a**) the IC50 and CR values of ZEN pAbs for mouse 5; (**b**) the IC50 and CR values of ZEN pAbs for mouse 2. The data were calculated from triplicate assays.

**Figure 6 toxins-14-00220-f006:**
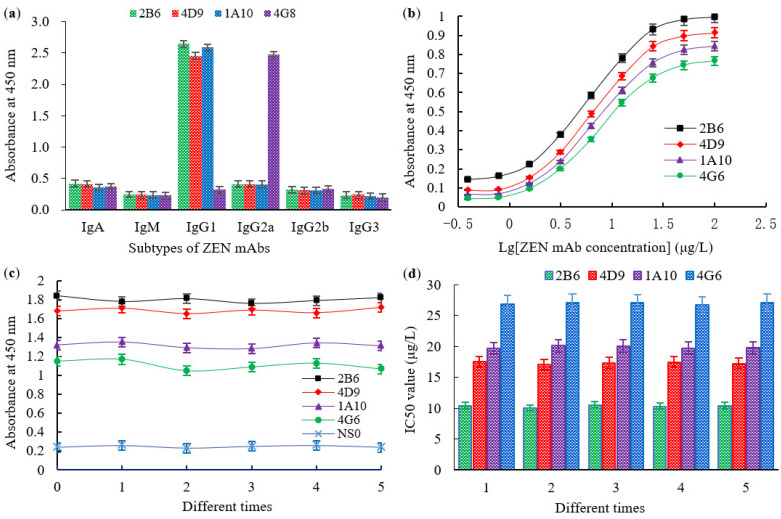
Characterization of ZEN mAbs obtained in this study: (**a**) analysis of the subtypes of ZEN mAbs 2B6, 4D9, 1A10, and 4G8; (**b**) affinity constant (Ka) of ZEN mAbs 2B6, 4D9, 1A10, and 4G8; (**c**) the titer values of ZEN mAbs 2B6, 4D9, 1A10, and 4G8 at different times; (**d**) the IC50 values of ZEN mAbs 2B6, 4D9, 1A10, and 4G8 at different times.

**Figure 7 toxins-14-00220-f007:**
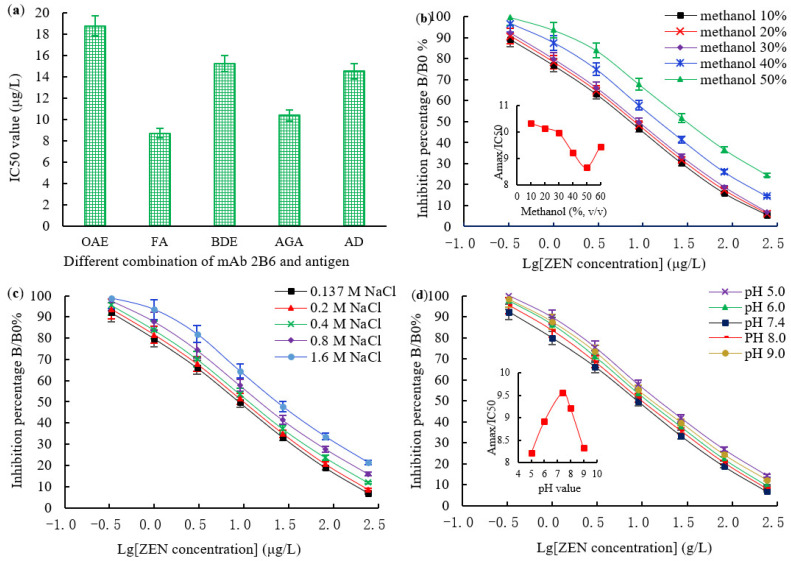
Optimization of icELISA experimental conditions and establishment of the icELISA standard curve: (**a**) selection of different combinations of ZEN mAb and antigen; (**b**) the effects of methanol contents on the sensitivity of the icELISA (insets indicate the fluctuations of Amax/IC50 as a function of methanol contents); (**c**) the effects of ionic strengths on the sensitivity of the icELISA (insets indicate the fluctuations of Amax/IC50 as a function of ionic strengths); (**d**) the effects of pH values on the sensitivity of the icELISA (insets indicate the fluctuations of Amax/IC50 as a function of pH values.

**Figure 8 toxins-14-00220-f008:**
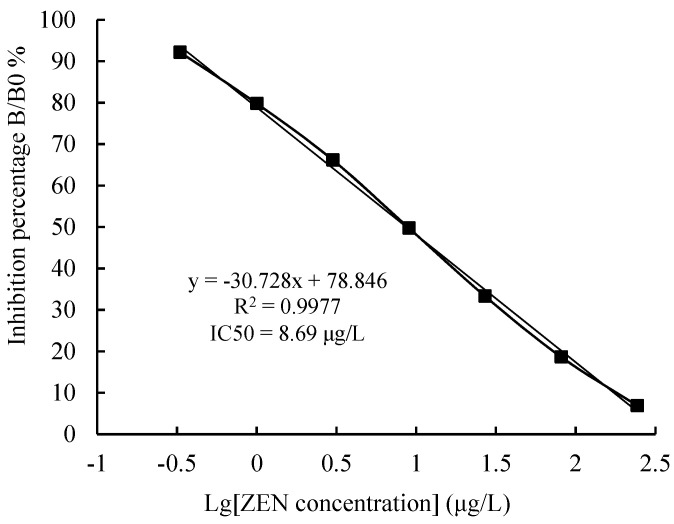
The standard curve of the icELISA for ZEN.

**Figure 9 toxins-14-00220-f009:**
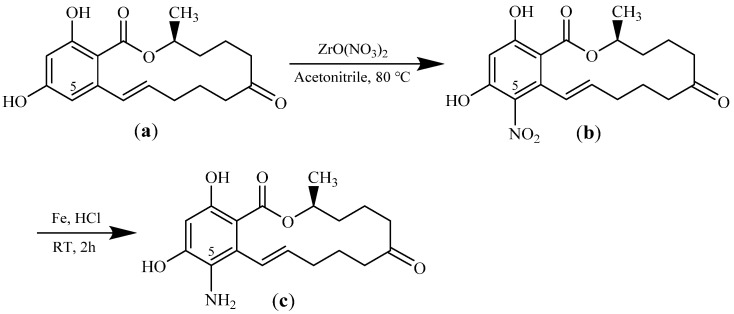
Synthesis route of hapten 5-NH_2_-ZEN: (**a**) zearalenone (ZEN); (**b**) 5-NO_2_-ZEN; (**c**) 5-NH_2_-ZEN.

**Figure 10 toxins-14-00220-f010:**
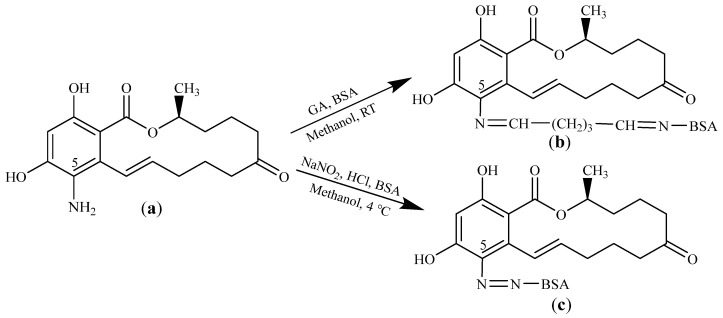
Synthesis route of immunogen ZEN-BSA via AGA and AD method: (**a**) zearalenone (ZEN); (**b**) ZEN-BSA (AGA); (**c**) ZEN-BSA (AD).

**Figure 11 toxins-14-00220-f011:**
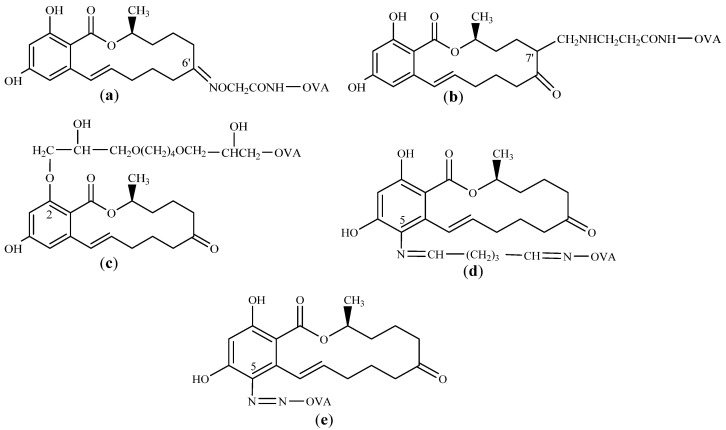
Synthesis routes of coating antigen ZEN-OVA via OAE, FA, BDE, AGA, and AD methods: (**a**) ZEN-OVA (OAE); (**b**) ZEN-OVA (FA); (**c**) ZEN-OVA (BDE); (**d**) ZEN-OVA (AGA); (**e**) ZEN-OVA (AD).

**Table 1 toxins-14-00220-t001:** The sensitivity (IC50 values) and specificity (CR values) of four mAbs against ZEN.

Compound	2B6		4D9		1A10		4G8	
	IC50 (μg/L) ^a,b^	CR (%) ^c^	IC50 (μg/L) ^a,b^	CR (%) ^c^	IC50 (μg/L) ^a,b^	CR (%) ^c^	IC50 (μg/L) ^a,b^	CR (%) ^c^
ZEN	10.38	100	17.23	100	19.87	100	27.05	100
α-ZAL	682.89	1.52	1057.06	1.63	296.13	6.71	379.92	7.12
β-ZAL	810.94	1.28	1276.30	1.35	299.25	6.64	390.33	6.93
α-ZOL	393.18	2.64	602.45	2.86	176.62	11.25	231.99	11.66
β-ZOL	567.21	1.83	857.21	2.01	243.51	8.16	317.49	8.52
ZON	243.09	4.27	353.07	4.88	103.12	19.12	132.86	20.36
Aflatoxin B1	>10,000	<1	>10,000	<1	>10,000	<1	>10,000	<1
Deoxynivalenol	>10,000	<1	>10,000	<1	>10,000	<1	>10,000	<1
T-2 toxin	>10,000	<1	>10,000	<1	>10,000	<1	>10,000	<1
Ochratoxin	>10,000	<1	>10,000	<1	>10,000	<1	>10,000	<1

Note. ^a^ The data were calculated from triplicate assays. The average coefficient of variation (CV) was below 10%. ^b^ The compound standard solution was prepared in 70% methanol-PBS (phosphate buffer saline, 0.01 M, pH 7.4) (7:3, *v*/*v*). ^c^ The data were calculated using the CR values of the mAbs against ZEN as 100%.

**Table 2 toxins-14-00220-t002:** Comparison of IC50 values and CR values of ZEN specific antibodies reported in previous literature.

References	ZEN Antibody	Coupling Method	Immunoassay Format	IC50 of ZEN (μg/L)	CR (%) ^a^				
					α-ZAL	β-ZAL	α-ZOL	β-ZOL	ZON
This study	ZEN mAb 2B6	AGA	icELISA ^b^	10.38	1.52	1.28	2.64	1.83	4.27
	ZEN mAb 1A10	AD	icELISA	19.87	6.71	6.64	11.25	8.16	19.12
Sun et al. (2014) [20]	ZEN mAb 4A3-F9	BDE	icELISA	1.115	3.854	1.709	2.499	2.800	53.121
Dong et al. (2018) [21]	ZEN mAb 6C2	OAE	icELISA	0.114	89.48	39.26	99.22	55.45	44.34
Burmistrova et al. (2009) [23]	ZEN mAb 2D8	OAE	dcELISA ^c^	0.8	69	<1	42	<1	22
Liu et al. (2015) [22]	ZEN mAb 7B2	OAE	BA-ELISA ^d^	0.18	46.7	39.2	60.5	24.7	59.5
Burkin et al. (2002) [24]	ZEN pAb	FA	icELISA	31.7	0.12	-	0.15	0.02	-
Gao et al. (2012) [25]	ZEN pAb	FA	icELISA	233.35	2.25	5.65	3.14	1.96	6.79
	ZEN mAb #	FA	icELISA	55.72	0.63	0.92	0.65	0.94	1.48
Teshima et al. (1990) [26]	ZEN mAb 7-1-144	AGA	icELISA	11.2	<0.1	<0.1	0.9	<0.1	4.0

Note. ^a^ The data were calculated using the CR values of ZEN as 100%. ^b^ icELISA: indirect competitive enzyme-linked immunosorbent assay. ^c^ dcELISA: direct competitive enzyme-linked immunosorbent assay. ^d^ BA-ELISA: biotin-streptavidin amplified enzyme-linked immunosorbent assay. - No data. #: unnamed.

**Table 3 toxins-14-00220-t003:** Accuracy and precision measurements of the icELISA for ZEN.

Samples	ZEN Spiked (μg/L)	Inner Batch			Among Batches		
		Measured Value (μg/L) ^a^	Recovery (%) ^b^	CV (%) ^a,c^	Measured Value (μg/L) ^a^	Recovery (%) ^b^	CV (%) ^a,c^
Maize meal ^d^	10	10.34 ± 1.13	103.4	13.7	10.21 ± 0.98	102.1	13.5
	20	19.34 ± 1.56	96.7	12.5	19.48 ± 1.36	97.4	12.4
	40	37.44 ± 2.21	93.5	11.3	37.68 ± 1.86	94.2	12.4
	80	72.16 ± 2.87	90.2	13.6	72.41 ± 2.11	90.5	11.8
Wheat meal ^e^	10	9.77 ± 1.42	97.7	12.9	9.81 ± 0.86	98.1	13.5
	20	18.82 ± 173	94.1	13.4	18.92 ± 1.28	94.6	13.3
	40	36.91 ± 2.43	92.3	12.6	37.24 ± 1.83	93.1	12.6
	80	71.84 ± 3.04	89.8	11.3	72.72 ± 2.41	90.9	11.7
Pig feed ^f^	10	10.67 ± 1.41	106.7	14.2	10.47 ± 1.22	104.7	14.6
	20	20.46 ± 1.87	102.3	13.8	19.74 ± 1.41	98.7	13.2
	40	38.64 ± 2.61	96.6	12.4	36.72 ± 1.74	91.8	12.7
	80	73.84 ± 3.56	92.3	11.5	71.44 ± 2.46	89.3	13.8

Note. ^a^ Measured values are presented as mean of six replicates. ^b^ recovery (%) = (concentration measured/concentration spiked) × 100. ^c^ CV indicates the coefficient of variation. ^d^ Peeled powder of common yellow maize. ^e^ Bran-removed powder of common red wheat. ^f^ Complete formula feed of young pigs.

**Table 4 toxins-14-00220-t004:** Comparison of the icELISA with HPLC-MS/MS for determination of ZEN in actual samples.

Samples	Sample Number	Positive Sample Number		Positive Rate (%)	Positive Sample Content Range (μg/L)		CV (%)
		icELISA ^a^	HPLC-MS/MS ^b^		icELISA ^a^	HPLC-MS/MS ^b^	
Maize meal	6	2	2	33.33	5.62–48.73	5.45–47.21	9.7
Wheat meal	6	1	1	16.67	23.51	22.84	11.2
Pig feed	6	2	2	33.33	21.42–64.36	20.78–62.81	10.1
Positive maize meal	2	2	2	100	24.62–66.35	23.15–36.42	9.8
Positive pig feed	2	2	2	100	37.22–71.54	36.38–71.69	10.6
Total	22	9	9	40.91	5.62–71.54	5.45–71.69	9.8–11.2

Note: ^a^ Results were determined using a heteroglogous icELISA format. ^b^ Results were determined via HPLC-MS/MS.

## Data Availability

Not applicable.

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
