# Peer review of "Development of a Highly Sensitive and Specific Monoclonal Antibody Based on Indirect Competitive Enzyme-Linked Immunosorbent Assay for the Determination of Zearalenone in Food and Feed Samples"

_toxins, 2022, doi:10.3390/toxins14030220_

Round 1

Reviewer 1 Report

MANUSCRIPT ID:  toxins- 1619990

Title: Development of a highly sensitive and specific monoclonal antibody based on indirect competitive enzyme-linked immune-sorbent assay for the determination of zearalenone in food and feed samples.

Submitted for publication to: Toxins

The aim of the present study is the development and the optimization of an enzyme-linked immunosorbent assay for the detection of ZEN in food and feed. Initially, a ZEN monoclonal antibody was synthesized and its characteristics (sensitivity, specificity) were assessed. Following, the optimal performance parameters of the assay were studied, selected and confirmed using HPLC-MS/MS in spiked samples.  

The experimental work is very well presented using a step-by-step approach and providing the necessary justification for each step, based on literature data or previous experimental work. Also, a comparative assessment with other published ZEN antibodies is provided. The manuscript is recommended to be published at its present form, taking into consideration the following notices.

Lines 56-62: The term “MRL” (abbreviation of Maximum Residue Limit) is used as a reference to the legal maximum level that is set for ZEN. The term MRL is generally used for the maximum limit of the pesticide residues. Instead, the term “maximum limit” is used for the food contaminants, such as ZEN (and other mycotoxins).

Lines 57-61: The legally established maximum limits of ZEN should be updated and verified. For example, according to the valid EU legislation, the maximum limit of ZEN for cereals and cereal products intended for human consumption is 75µg/kg.

Lines 82-84: Unmeaning repeat of the exact phrase in Lines 200-202.

Lines 508-512: It is described that the blank samples (2g each) is spiked with four different concentrations (10, 20, 40, 80 μg/L) of ZEN.  However, since the total volume (ml) of the ZEN solution that was used to spike the 2g blank sample is not mentioned, the performance of the test at the concentrations close to the legal maximum limits (expressed in µg/kg units) cannot be assessed. This information could be of particular utility and importance showing its potential to be used as rapid analysis tool for the separation of the contaminated batches.

Author Response

  1. Lines 56-62: The term “MRL” (abbreviation of Maximum Residue Limit) is used as a reference to the legal maximum level that is set for ZEN. The term MRL is generally used for the maximum limit of the pesticide residues. Instead, the term “maximum limit” is used for the food contaminants, such as ZEN (and other mycotoxins).

Thanks for your valuable advice. According to your suggestion, we have studied and reviewed the relevant literature, and indeed should not use "Maximum Residue Limit (MRL)", should use "maximum limits", we have made changes, marked in yellow, see lines 59, 61, 63, and 65.

  1. Lines 57-61: The legally established maximum limits of ZEN should be updated and verified. For example, according to the valid EU legislation, the maximum limit of ZEN for cereals and cereal products intended for human consumption is 75µg/kg.

We are sorry for our careless, and we have corrected them and marked it in yellow on lines 61-65.

  1. Lines 82-84: Unmeaning repeat of the exact phrase in Lines 200-202.

Thanks for your valuable advice, according to your suggestion, we have modified and rewrote sentence,marked it in yellow on lines 213-218.

  1. Lines 508-512: It is described that the blank samples (2g each) is spiked with four different concentrations (10, 20, 40, 80 μg/L) of ZEN.  However, since the total volume (ml) of the ZEN solution that was used to spike the 2g blank sample is not mentioned, the performance of the test at the concentrations close to the legal maximum limits (expressed in µg/kg units) cannot be assessed. This information could be of particular utility and importance showing its potential to be used as rapid analysis tool for the separation of the contaminated batches.

Thank you for your valuable guidance .We are very sorry for us not explaining clearly. We have made changes carefully. First, in the actual experiment, the ZEN standard stock solution was diluted with 70% methanol-PBS (7:3, v/v) at a concentration of 2000 μg/L. Modified sections are marked in yellow on lines 358-360. Second, the ZEN standard stock solution was diluted with 70% methanol-PBS (7:3, v/v) to different ZEN concentrations (0.33, 1.0, 3.0, 9.0, 27.0, 81.0, and 243.0.0 μg/L), to establish a standard curve of icELISA. The modified parts are marked in yellow on line 535. The third is sample treatment. The volume of the sample extract after processing was 100 mL. Modified sections are marked in yellow on lines 373-376. The fourth is the treatment of spiked samples. The four blank samples were spiked with ZEN stock solution with a concentration of 2000 μg/L, the final volume was 2.0 mL, and the four different ZEN concentrations were (10, 20, 40, 80 μg/L). Modified sections are marked in yellow on lines 550-555. We hope you can accept our explanations and revisions.

Reviewer 2 Report

I suggest few minor corrections

In Fig 2a on horizontal ax put breaks between 220240260280300320340360380400420440

line 150 , the correct value is probably 6,4x 103

line 152 , the correct value is probably 3,2x 103

line 165-166 delete , 2.2. Figures, Tables and Schemes

in Table 3 (page 10) make a clear delimitation between categories  Maize meal, Wheat meal and Pig Feed

Author Response

1.I suggest few minor corrections. In Fig 2a on horizontal ax put breaks between 220240260280300320340360380400420440.

Thanks for your valuable advice. According to your suggestion, we have revised Figure 2a and 2b, mainly by adjusting the scale of the axes to make the different curves easier to identify. The modified parts are marked in grey on line 153. 

2.Line 150 , the correct value is probably 6,4x 103

Thank you for your kind reminder. The correct value is 6.4×103, due to a formatting superscript issue. We have corrected it, and marked it in gray on line 160.

3.Line 152 , the correct value is probably 3,2x 103

Thank you for your kind reminder. The correct value is 3.2×103, due to a formatting superscript issue. We have corrected it, and marked it in gray on line 162.

4.line 165-166 delete, 2.2. Figures, Tables and Schemes

We are sorry for our careless, and we have deleted the excrescent parts, and marked them in gray on lines 175-176.

5.In Table 3 (page 10) make a clear delimitation between categories  Maize meal, Wheat meal and Pig Feed.

Thanks for your valuable advice. According to your suggestion, we have made a clear delimitation for four samples, and marked them in gray on lines 308-309.

Reviewer 3 Report

Journal: Toxins

Manuscript ID: Toxins-1619990

Title: Development of a highly sensitive and specific monoclonal antibody based on indirect competitive enzyme-linked immuno-sorbent assay for the determination of zearalenone in food and feed samples

Comments to the Author  

Overall, the manuscript is well written. The experimental design and analysis have no major flaws. The authors gather the possible significant information for providing a conclusion. I am thinking that the results from this research are very useful when it can apply in the practical production. In my opinion, this manuscript can be acceptable for publication.

Author Response

1.Overall, the manuscript is well written. The experimental design and analysis have no major flaws. The authors gather the possible significant information for providing a conclusion. I am thinking that the results from this research are very useful when it can apply in the practical production. In my opinion, this manuscript can be acceptable for publication.

Thank you for your affirmation and encouragement to the manuscript, we will definitely check it carefully and continue to work hard.

Reviewer 4 Report

The work “Development of a highly sensitive and specific monoclonal an-tibody based on indirect competitive enzyme-linked immuno-sorbent assay for the determination of zearalenone in food and feed samples” touches upon the important process of obtaining antibodies with an increase in specificity for zearalenone. Obtaining new antibodies for immunochemical analysis is an important step in the process of creating a test method. Validation of the method confirms its sensitivity even in real samples. I have couple of question:

- Line 239-241. “The chessboard titration results indicated that the op- 239 timal concentrations of the coating antigen, mAb 2B6 and GaMIgG-HRP were 2 μg/mL, 240 0.5 μg/mL (1:10000), and 0.6 μg/mL (1:2000), respectively (data not shown).” From what concentrations was the choice made? On the basis of what was the optimal concentration chosen?

- Are inELISA and icELISA suitable for work in industry? What is the limit of detection of zearalenone for this type of product set by law?

Minor comments

  • Figure 2 A, please change scale step. It is difficult to read
  • Last reference looks different; be careful during next data submission

Author Response

  1. Line 239-241. “The chessboard titration results indicated that the op- 239 timal concentrations of the coating antigen, mAb 2B6 and GaMIgG-HRP were 2 μg/mL, 240 0.5 μg/mL (1:10000), and 0.6 μg/mL (1:2000), respectively (data not shown).” From what concentrations was the choice made? On the basis of what was the optimal concentration chosen?

Thank you for your valuable advice,and we are very sorry for our unclear explanation. The manuscript described the determination of optimal working concentrations (on lines 528-531) of the coating antigens ZEN-OVA, ZEN mAb, and GaMIgG-HRP by checkerboard titration in section "4.6. Development and optimization of icELISA", but in "2.3. icELISA optimization and establishment of icELISA standard curve" section was not explained clearly, we have made modifications and supplements, marked in green in lines 259-264. At the same time, due to the large amount of experimental data for the checkerboard titration method, a hint is made in the manuscript (data not shown). We hope you can accept our interpretation.

  1. Are inELISA and icELISA suitable for work in industry? What is the limit of detection of zearalenone for this type of product set by law?

We are very sorry for our unclear explanation. The inELISA is mainly used to determine the titers of antibodies produced in serum after immunization, which is mainly used for clinical diagnosis. The icELISA is mainly used to determine the content of pollutants such as viruses, bacteria, drugs and mycotoxins in positive samples. It can be used not only as a clinical diagnostic reagent, but also as a food safety detection reagent. In the field of food safety detection, it is mainly used for the detection of pollutants in agricultural products, as well as the detection of food industrial products and feed industrial products, but it is generally not used in other industrial fields.

As for the detection of limit (LOD) of zearalenone in food stipulated by law, the manuscript lists some examples in the "1. Introduction" section, such as the European Union, Italy, Australia and China. However, in different countries and regions, there are different legal provisions on the maximum limits of different food and feed products. For the performance requirements such as detection limit and detection range of products tested by icELISA kit, AOAC took the detection method of icELISA kit as ZEN legal detection method (AOAC official method 994.01, zearalenone in corn, wheat, and feed Enzymelinked immunosorbent (agri screen) method first action 1994, final action 1997), and the detection of limit was 800 μg/L. In 2004, China took the detection method of icELISA kit as ZEN legal detection method (GB/T 19540-2004 Determination of Zearalenone in Feed), but there was no strict detection of limit, and the general requirement was 1000 μg/L.

3.Figure 2 A, please change scale step. It is difficult to read.

Thanks for your valuable advice. According to your suggestion, we have revised Figure 2a and 2b, mainly by adjusting the scale of the axes to make the different curves easier to identify. The modified parts are marked in green on line 153.

4.Last reference looks different; be careful during next data submission.

Thank you for your encouragement and kind reminder. We will check the manuscript carefully.
